## [Reviewer comments · Royal Society Open Science]

Review History

RSOS-201200.R0 (Original submission)

Review form: Reviewer 1 (Peter Holland)

Is the manuscript scientifically sound in its present form?

Yes

Are the interpretations and conclusions justified by the results?

Yes

Is the language acceptable?

Yes

Do you have any ethical concerns with this paper?

No

Have you any concerns about statistical analyses in this paper?

No

Recommendation?

Accept with minor revision (please list in comments)

Comments to the Author(s)

Main comments

This paper presents new data relevant to understanding the molecular basis of an interesting evolutionary transition from ammonia production to urea production in a genus of fish living in alkaline lakes. The authors conclude that this occurred by convergent evolution with tetrapods and present several lines of evidence indicating the types of molecular change that occurred. Notably, the alkaline lake-dwelling fish have acquired amino acid changes in an enzyme (CPS III) which is a paralogue of the enzyme used by terrestrial vertebrates for the same role (CPS I) and these amino acid changes are predicted here to alter substrate binding. This convergent evolution in enzyme activity is not verified experimentally but the protein sequence changes are highly suggestive. In addition, the gene seems to have changed site of expression, allowing nitrogenous waste handling in active muscle tissues; the authors show this clearly by RT-PCR and in situ hybridisation. Finally, the authors suggest that the expression change may have been driven by promoter sequence change, for which DNA sequence evidence is given but not experimental evidence. Together, the sequence and expression data are clear, and although not tested by studies of enzyme activity or transcription factor binding studies, they are mutually supportive and form a robust study.

There are two small issues that needs clarifying but these should be straightforward. First, the promoter sequence should be given. Second, no evidence is presented that the gene being studied in *Alcolapia* is CPS III. Perhaps this could be rectified by a phylogenetic tree of complete CPS I, CPS II and CPS III amino acid sequences from a range of vertebrates; the expectation is that *Alcolapia* CPS III would be nested within a clade of CPS III genes (convergent changes in a few amino acids are not expected to disrupt this). I would recommend this should be done using amino acid sequences, a maximum likelihood or Bayesian method, and a model of substitution incorporating gamma distributed rates to account for within-site rate heterogeneity; branch lengths should be displayed.

Typographical or suggestions for clarity

Alcolapia is considered a genus by some authors; perhaps the second sentence of the abstract - or Line 39 in Introduction - should note this?

Line 26 "teleosts"

Line 66 "In terrestrial vertebrates.."

Line 67 "generate"

Line 69 "in the evolution of amphibian species" is ambiguous. Does this mean in the clade comprising living amphibians only? Or does it mean within (a subset of) the living amphibians? Or does it mean on the stem lineage of the clade comprising living amphibians plus amniotes (i.e. the stem lineage of living tetrapods)? I think the implication is the last of these, but I am not sure. The same applies to lines 252 and 257. In each case, this phrasing of an evolutionary hypothesis needs to be made clear in terms of which clade is being referred to.

Line 97-98 "aquatics habitat" Rephrasing may make clearer

Line 98 degree symbol

Line 102 manufacturers

Lines 108-114. Too much unnecessary detail is given for regular day-to-day subcloning. All that is needed for the reader is length of insert and statement that the probe was made by in vitro transcription from linearised pGEM®-T Easy plasmid.

Line 120 Components of hybridisation solution would be useful information (not just 50% formamide)

Line 125-126. Primer sequences need to be given.

Line 126 The deduced coding sequences (nucleotide sequence) for *A. alcalica* and *A. grahami* CPSIII need to be available. The paper gives two accession numbers but these do not seem 'live'. This needs correcting so readers can use the data. Also, it should be specified these are NCBI accession numbers.

Line 130. I note that the 3.5kb region 5' to CPS III was obtained from an unpublished genome sequence. I think it is necessary to give the 3.5 kb DNA sequence e.g. via an NCBI accession, to enable readers to conduct their own analyses of the promoter.

Line 131 delete word "was"

Line 145 and Figure 1: line 29, seems inaccurate to refer to LB (limb buds) in fish; perhaps PFB (pectoral fin bud) may be more accurate?

Line 151 insert word "putative" before "paired E-box"

Line 182 The term "worsening" indicates a value from some perspective; may be better to rephrase to 'changing'?

Line 200 semi-colon should be before 'for example', not after

Line 256-257 This sentence does not make sense to me and needs clarification. There was no "evolutionary transition" from "CPS III to CPS I" so what is being discussed in this sentence?

Line 262 adaptations

Review form: Reviewer 2 (Tetsuya Nakamura)

Is the manuscript scientifically sound in its present form?

Yes

Are the interpretations and conclusions justified by the results?

Yes

Is the language acceptable?

Yes

Do you have any ethical concerns with this paper?

No

Have you any concerns about statistical analyses in this paper?

No

Recommendation?

Accept as is

Comments to the Author(s)

The authors extensively answered my concerns and the manuscript is qualified to be published.

Decision letter (RSOS-201200.R0)

Dear Mr White,

On behalf of the Editors, we are pleased to inform you that your Manuscript RSOS-201200 "Adaptation of the carbamoyl-phosphate synthetase (CPS) enzyme in an extremophile fish" has been accepted for publication in Royal Society Open Science subject to minor revision in accordance with the referees' reports. Please find the referees' comments along with any feedback from the Editors below my signature.

We invite you to respond to the comments and revise your manuscript. Below the referees' and Editors' comments (where applicable) we provide additional requirements. Final acceptance of

your manuscript is dependent on these requirements being met. We provide guidance below to help you prepare your revision.

Please submit your revised manuscript and required files (see below) no later than 7 days from today's (ie 09-Sep-2020) date. Note: the ScholarOne system will 'lock' if submission of the revision is attempted 7 or more days after the deadline. If you do not think you will be able to meet this deadline please contact the editorial office immediately.

on behalf of Dr Rees Kassen (Associate Editor) and Kevin Padian (Subject Editor)
openscience@royalsociety.org

Associate Editor Comments to Author (Dr Rees Kassen):

The revised manuscript outlines an exceptional case of nitrogen excretion in African fish species occupying a high alkaline lake. The main results concern the physiology and putative genetic changes associated with urea, rather than the more usual ammonia, excretion. Two referees have reviewed the manuscript and both are positive in their assessments. In my reading of the manuscript I do not see any reason to question the referee's assessments, though I concur with the more critical reviewer that the authors must provide the DNA sequences (or make these accessible) and would ideally provide additional data supporting the identity and, ideally, phylogenetic position of CPS III. This reviewer also provides some additional comments related to readability that are helpful in revising the manuscript. Provided these changes are made, I recommend accepting the manuscript for publication.

Reviewer comments to Author:

Reviewer: 1
Comments to the Author(s)

Main comments

This paper presents new data relevant to understanding the molecular basis of an interesting evolutionary transition from ammonia production to urea production in a genus of fish living in alkaline lakes. The authors conclude that this occurred by convergent evolution with tetrapods and present several lines of evidence indicating the types of molecular change that occurred. Notably, the alkaline lake-dwelling fish have acquired amino acid changes in an enzyme (CPS III) which is a paralogue of the enzyme used by terrestrial vertebrates for the same role (CPS I) and these amino acid changes are predicted here to alter substrate binding. This convergent evolution in enzyme activity is not verified experimentally but the protein sequence changes are highly suggestive. In addition, the gene seems to have changed site of expression, allowing nitrogenous

waste handling in active muscle tissues; the authors show this clearly by RT-PCR and in situ hybridisation. Finally, the authors suggest that the expression change may have been driven by promoter sequence change, for which DNA sequence evidence is given but not experimental evidence. Together, the sequence and expression data are clear, and although not tested by studies of enzyme activity or transcription factor binding studies, they are mutually supportive and form a robust study.

There are two small issues that needs clarifying but these should be straightforward. First, the promoter sequence should be given. Second, no evidence is presented that the gene being studied in *Alcolapia* is CPS III. Perhaps this could be rectified by a phylogenetic tree of complete CPS I, CPS II and CPS III amino acid sequences from a range of vertebrates; the expectation is that *Alcolapia* CPS III would be nested within a clade of CPS III genes (convergent changes in a few amino acids are not expected to disrupt this). I would recommend this should be done using amino acid sequences, a maximum likelihood or Bayesian method, and a model of substitution incorporating gamma distributed rates to account for within-site rate heterogeneity; branch lengths should be displayed.

Typographical or suggestions for clarity

Alcolapia is considered a genus by some authors; perhaps the second sentence of the abstract - or Line 39 in Introduction - should note this?

Line 26 "teleosts"

Line 66 "In terrestrial vertebrates.."

Line 67 "generate"

Line 69 "in the evolution of amphibian species" is ambiguous. Does this mean in the clade comprising living amphibians only? Or does it mean within (a subset of) the living amphibians? Or does it mean on the stem lineage of the clade comprising living amphibians plus amniotes (i.e. the stem lineage of living tetrapods)? I think the implication is the last of these, but I am not sure. The same applies to lines 252 and 257. In each case, this phrasing of an evolutionary hypothesis needs to be made clear in terms of which clade is being referred to.

Line 97-98 "aquatics habitat" Rephrasing may make clearer

Line 98 degree symbol

Line 102 manufacturers

Lines 108-114. Too much unnecessary detail is given for regular day-to-day subcloning. All that is needed for the reader is length of insert and statement that the probe was made by in vitro transcription from linearised pGEM®-T Easy plasmid.

Line 120 Components of hybridisation solution would be useful information (not just 50% formamide)

Line 125-126. Primer sequences need to be given.

Line 126 The deduced coding sequences (nucleotide sequence) for *A. alcalica* and *A. grahami* CPSIII need to be available. The paper gives two accession numbers but these do not seem 'live'. This needs correcting so readers can use the data. Also, it should be specified these are NCBI accession numbers.

Line 130. I note that the 3.5kb region 5' to CPS III was obtained form an unpublished genome sequence. I think it is necessary to give the 3.5 kb DNA sequence e.g. via an NCBI accession, to enable readers to conduct their own analyses of the promoter.

Line 131 delete word "was"

Line 145 and Figure 1: line 29, seems inaccurate to refer to LB (limb buds) in fish; perhaps PFB (pectoral fin bud) may be more accurate?

Line 151 insert word "putative" before "paired E-box"

Line 182 The term "worsening" indicates a value from some perspective; may be better to rephrase to 'changing'?

Line 200 semi-colon should be before 'for example', not after

Line 256-257 This sentence does not make sense to me and needs clarification. There was no "evolutionary transition" from "CPS III to CPS I" so what is being discussed in this sentence?

Line 262 adaptations

Reviewer: 2
 Comments to the Author(s)

The authors extensively answered my concerns and the manuscript is qualified to be published.

===PREPARING YOUR MANUSCRIPT===

===PREPARING YOUR REVISION IN SCHOLARONE===

-- If you have uploaded ESM files, please ensure you follow the guidance at <https://royalsociety.org/journals/authors/author-guidelines/#supplementary-material> to include a suitable title and informative caption. An example of appropriate titling and captioning may be found at https://figshare.com/articles/Table_S2_from_Is_there_a_trade-off_between_peak_performance_and_performance_breadth_across_temperatures_for_aerobic_scops_in_teleost_fishes_/3843624.

Author's Response to Decision Letter for (RSOS-201200.R0)

See Appendix A.

Decision letter (RSOS-201200.R1)

Dear Mr White,

It is a pleasure to accept your manuscript entitled "Adaptation of the carbamoyl-phosphate synthetase (CPS) enzyme in an extremophile fish" in its current form for publication in Royal Society Open Science.

on behalf of Dr Rees Kassen (Associate Editor) and Kevin Padian (Subject Editor)
openscience@royalsociety.org

Appendix A

We are very grateful to the editor and reviewers for their positive comments on this study. We have edited this manuscript in accordance with their main concerns and believe that these changes have improved our manuscript. From these comments there were three main concerns that need to be addressed and we believe that we have now corrected all of these. Firstly, it was suggested that the CPS III upstream region discussed in the paper be uploaded to NCBI for ease of access, this has now been done. Secondly, the reviewer wished for proof that the CPS III genes which we sequenced from *Alcolapia* species were definitely CPS III. To do this we produced, and provide as supplementary data, a phylogeny showing that the genes from *Alcolapia* clade with other fish CPS III and with tetrapod CPS I. Thirdly, from the comments about understanding of the manuscript we felt it was not made clear that CPS III in fish and CPS I in tetrapods are syntenic orthologues. We have clarified this with a further statement in the introduction of the manuscript for ease of understanding.

Finally below are detailed changes addressing particular concerns from reviewers:

Alcolapia is considered a genus by some authors; perhaps the second sentence of the abstract - or Line 39 in Introduction - should note this?

- This has been added to the introduction as suggested for clarification about the species being discussed

Line 26 “teleosts”

- Change incorporated

Line 66 “In terrestrial vertebrates..”

- Change incorporated

Line 67 “generate”

- Change incorporated

Line 69 “in the evolution of amphibian species” is ambiguous. Does this mean in the clade comprising living amphibians only? Or does it mean within (a subset of) the living amphibians? Or does it mean on the stem lineage of the clade comprising living amphibians plus amniotes (i.e. the stem lineage of living tetrapods)? I think the implication is the last of these, but I am not sure. The same applies to lines 252 and 257. In each case, this phrasing of an evolutionary hypothesis needs to be made clear in terms of which clade is being referred to.

- Sentences at lines 69, 252 and 257 have been edited to better clarify its meaning about when the change in function is believed to have occurred

Line 97-98 “aquatics habitat” Rephrasing may make clearer

- Sentence changed for clarity

Line 98 degree symbol

- Change incorporated

Line 102 manufacturers

- Change incorporated

Lines 108-114. Too much unnecessary detail is given for regular day-to-day subcloning. All that is needed for the reader is length of insert and statement that the probe was made by in vitro transcription from linearised pGEM[®]-T Easy plasmid.

- Method for cloning made more concise

Line 120 Components of hybridisation solution would be useful information (not just 50% formamide)

- Method expanded to include reagent composition

Line 125-126. Primer sequences need to be given.

- Table of primers included as supplementary data

Line 126 The deduced coding sequences (nucleotide sequence) for *A. alcalica* and *A. grahami* CPSIII need to be available. The paper gives two accession numbers but these do not seem 'live'. This needs correcting so readers can use the data. Also, it should be specified these are NCBI accession numbers.

- We have specified that these are NCBI accession numbers and will become live on the 1st of October. We have included the sequences as supplemental data in the meantime

Line 130. I note that the 3.5kb region 5' to CPS III was obtained from an unpublished genome sequence. I think it is necessary to give the 3.5 kb DNA sequence e.g. via an NCBI accession, to enable readers to conduct their own analyses of the promoter.

- This has now been submitted to NCBI and accession number included in the manuscript

Line 131 delete word "was"

- Change incorporated

Line 145 and Figure 1: line 29, seems inaccurate to refer to LB (limb buds) in fish; perhaps PFB (pectoral fin bud) may be more accurate?

- We have changed the figure and text to now refer to appendage as pectoral fin bud

Line 151 insert word "putative" before "paired E-box"

- Change incorporated

Line 182 The term "worsening" indicates a value from some perspective; may be better to rephrase to 'changing'?

- Change incorporated

Line 200 semi-colon should be before 'for example', not after

- Change incorporated

Line 256-257 This sentence does not make sense to me and needs clarification. There was no "evolutionary transition" from "CPS III to CPS I" so what is being discussed in this sentence?

- We have clarified in the introduction that CPS III and I are orthologues and that the reason for the difference in name is their biochemical properties but still remain the same gene. We believe that this incorporation of information addresses the reviewers comment here

Line 262 adaptations

- Change incorporated